# Genetics of Body Fat Distribution: Comparative Analyses in Populations with European, Asian and African Ancestries

**DOI:** 10.3390/genes12060841

**Published:** 2021-05-29

**Authors:** Chang Sun, Peter Kovacs, Esther Guiu-Jurado

**Affiliations:** 1Medical Department III–Endocrinology, Nephrology, Rheumatology, University of Leipzig Medical Center, 04103 Leipzig, Germany; sun.chang@medizin.uni-leipzig.de (C.S.); Peter.kovacs@medizin.uni-leipzig.de (P.K.); 2Deutsches Zentrum für Diabetesforschung, 85764 Neuherberg, Germany

**Keywords:** body fat distribution, genetics, GWAS

## Abstract

Preferential fat accumulation in visceral vs. subcutaneous depots makes obese individuals more prone to metabolic complications. Body fat distribution (FD) is regulated by genetics. FD patterns vary across ethnic groups independent of obesity. Asians have more and Africans have less visceral fat compared with Europeans. Consequently, Asians tend to be more susceptible to type 2 diabetes even with lower BMIs when compared with Europeans. To date, genome-wide association studies (GWAS) have identified more than 460 loci related to FD traits. However, the majority of these data were generated in European populations. In this review, we aimed to summarize recent advances in FD genetics with a focus on comparisons between European and non-European populations (Asians and Africans). We therefore not only compared FD-related susceptibility loci identified in three ethnicities but also discussed whether known genetic variants might explain the FD pattern heterogeneity across different ancestries. Moreover, we describe several novel candidate genes potentially regulating FD, including *NID2*, *HECTD4* and *GNAS*, identified in studies with Asian populations. It is of note that in agreement with current knowledge, most of the proposed FD candidate genes found in Asians belong to the group of developmental genes.

## 1. Introduction

### 1.1. Body Fat Distribution in Metabolic Diseases

It is well-known that an obesity pandemic has reached most corners of the world, becoming one of the most severe and unavoidable public health problems of this century. Obesity substantially increases the risk of certain comorbidities such as type 2 diabetes mellitus (T2D), hypertension, cardiovascular disease and metabolic syndrome and many more [1]. However, obesity, per se, does not necessarily lead to these pathologies. Numerous studies presented a unique subgroup of overweight and obese individuals who appear to be protected from obesity-related metabolic disturbances [2,3,4]. These individuals who have normal metabolic features despite increased adiposity are considered to be metabolically healthy obese (MHO). Although this may suggest that obesity is a heterogeneous disorder rendering a variable risk for individuals, it does not mean that body weight gain is associated with a healthy pattern. It is more likely that MHO is only a transitional period [5], as has been shown by a number of meta-analyses that compared metabolically healthy normal-weight individuals with MHO individuals with obesity [5,6,7,8,9]. In summary, even though metabolically healthy, the obese group was still at greater risk of adverse long-term outcomes, including T2D [6], cardiovascular events [7], metabolic abnormalities [8] and dyslipidemia [9]. Moreover, a prospective study including Japanese Americans revealed that the independent risk factor of the transition from MHO to metabolically unhealthy obese (MUO) is the accumulation of visceral fat [10]. The MUO is usually classified as an android (visceral or upper-body abdominal region) or gynoid (subcutaneous or lower-body gluteofemoral region) obesity [11]. Indeed, adverse body fat distribution as seen in visceral obesity (body fat deposited preferentially in visceral depots) is the main metabolic and cardiovascular risk predictor in obesity [12]. Even after accounting for body mass index (BMI), visceral fat depots increase the risk of having T2D [13], cardiovascular disease [14] and all-cause mortality [15]. In two heart cohort studies (Framingham and Jackson Heart Study), the size of the abdominal visceral adipose tissue was more strongly associated with adverse cardiometabolic risk factors than the volume of the subcutaneous adipose tissue [14,16].

### 1.2. Measures of Body Fat Distribution

Waist circumference (WC), waist–hip ratio (WHR) and WHR adjusted for BMI (WHRadjBMI) are widely anthropometric parameters used to determine regional FD. Compared to more accurate magnetic resonance imaging (MRI) and computerized tomography (CT), these measurement indexes are easy to collect in large-scale applications aimed at evaluating body fat distribution.

WC as a proxy of abdominal obesity is a more accurate parameter to predict the risk of T2D than BMI and has also been shown to be associated with cardiometabolic disease [17]. WHR as an indicator of FD is linked to the risk of T2D and coronary disease independently of BMI [18]. These widely used anthropometric parameters have been applied to identify and assess the genetic determinants of FD in GWAS [19,20,21,22,23,24,25,26,27,28] and other genetics studies (e.g., fine mapping) [29,30].

### 1.3. Genetic and Non-Genetic Determinants of Body Fat Distribution

The main factors governing fat distribution are sex [31,32], genetics [19,20,24,33,34,35,36], epigenetics [37] and environment [38], such as a sedentary lifestyle and high sugar food intake. Although the predisposing factors of FD are heterogeneous, genetics appears to be a key determinant of FD as represented by heritability estimates ranging from 22% to 61% even after accounting for obesity [39]. Pérusse et al. estimated in a Quebec family study that genetics accounted for 56% of abdominal visceral fat, but only 42% of subcutaneous fat [40]. In a Danish twin study, total fat heritability was estimated between 83% and 86%, and between 71% and 85% for regional fat (trunk, lower-body, trunk/lower-body) [41]. Notably, the measured heritability depends on the environmental variance in the study population, which if it is low, can lead to its overestimation [42]. It has been well acknowledged that independent of obesity, FD patterns differ among populations. For example, Wang et al. showed that Asians have lower BMIs but higher percentages of total body fat than Europeans [43]. Under the same corresponding obesity categories, Chinese and South Asians have more body fat than Caucasians [43,44,45]. A multicultural community health assessment study showed that compared with Europeans, Chinese and South Asians have higher abdominal fat accumulation, especially in visceral adipose tissue [46]. Additionally, Africans have relatively more subcutaneous adipose tissue and less visceral fat tissues than European populations, independent of sex [47,48] and age [49]. The correlation between increased visceral fat and T2D risk may explain why Asians are more likely to suffer from T2D despite having lower BMIs [50]. By comparing with non-Hispanic whites and blacks, Asians have a higher risk of hypertension and T2D [51]. Specifically, for each unit increase in BMI, Asians had a 15% increase in the odds of diabetes, which is higher than an 11% increase among non-Hispanic whites, a 7% increase among blacks, and an 8% increase among Hispanics [51]. On the basis of these multi-populational FD studies and considering the adipose tissue expandability hypothesis [52], it might be speculated that Asians have a specific mechanism that prioritizes visceral fat accumulation, or that the subcutaneous fat expansion capacity of Asians is not sufficient to support the growing high-calorie intake and reduced physical activity lifestyle, leading to excessive lipid accumulation in visceral or ectopic adipocytes, and therefore ultimately resulting in an increased risk of developing metabolic disease. Furthermore, this is most likely potentiated by the interaction between genetics and environmental factors.

Within the last decade, hundreds of genetic risk loci of WHR and WC have been identified in GWAS, making remarkable progress in our knowledge about the genetics of fat distribution. However, GWAS may overestimate the effects of these susceptibility loci on traits or disease because the approach to assess ancestry variables in GWAS is often based on self-reported information. Considering the character of the approach, there may be a missing influence of the residual population, which is important for complex traits driven by a large number of variants of small effects. It may influence comparisons of complex traits between populations and overestimate polygenic adaptation due to residual population stratification, although it does not question the validity of the discovered genome-wide significant loci [53]. Additionally, currently established methods based on European ancestry samples such as Genetic Risk Score (GRS) or Polygenic Risk Score (PRS), which are individual-level metrics of genetic risk for human complex traits in unrelated individuals, have proposed that the predictive performance from studies in European ancestries is lower in non-European populations [54]. Indeed, although most GWAS studies have so far been conducted in European populations, their findings could most likely not be extrapolated to other populations. As it has been shown in our previous genetics of obesity review, almost 60% of known loci replication failed in the East Asian population [55]. To date, several GWAS of FD were successfully conducted in Asian populations, including individuals of diverse Asian ancestries (e.g., South Asians and East Asians). However, most findings from European studies were not able to be replicated in Asian cohorts or African cohorts. In this review, we aimed to summarize recent advances in the genetics of fat distribution by accounting for differences among populations. Therefore, we elaborated on FD susceptibility loci identified in European, Asian and African populations and discussed the potential role of known genetic variants in ancestry-related differences in FD patterns. We would like to point out that, although the term ’Caucasian’ appears to be outdated and potentially also refers to Africans and Asians, in this review, the respective term solely represents a group that includes only European and American Caucasians.

### 1.4. The Role of Ancestry in Genetic Studies

The majority of current genetic studies are based on self-assessed ancestries. Despite this approach being less intrusive for the study participants than collecting explicit data, when applied to disease risk assessment—particularly complex traits driven by a large number of variants—it proved less reliable compared with explicit genetic information on subjects’ descents. Recent studies indicate that the effects of variants from GWAS on complex traits may vary among populations despite p values reaching genome-wide statistical significance as has been shown, e.g., for height [53,56]. Likewise, polygenic scores derived from studies in cohorts of European ancestry have been shown to be less reliable in their predictive performance in populations of non-European ancestries [54].

Although common variants are strongly shared among populations, their frequencies often differ substantially. Moreover, despite the human genome being identical in each individual, genetic drift, natural selection and de novo mutations have led to a slight genetic difference between populations during migrations and in the course of evolution. Ancestry diversity is common in modern populations [57], which originated in Africa and spread to all other regions in the world. Europeans and Asians are descended from a small group of individuals who migrated out of Africa [58]. However, the number, geographic origin, migratory routes and the timing of major dispersals remain elusive. According to the currently available data, modern humans dispersed out of Africa 50 kyr–100 kyr ago northward or southward, while the migration separated immediately into two waves. One wave led ultimately to the founding of Australasia and New Guinea and the other contributed to the ancestry of present-day mainland Eurasians [59]. Europeans and Asians have undergone genomic mixture across the Eurasian steppes [60,61]. However, even if common ancestry is shared between different populations, it remains diverse due to long-term changes of the respective gene pools which have been shaped by the specific environment people were living in. With the exception of the shared steppe ancestry of Europeans in present-day Asia, evolutionary DNA analyses indicated that South, Southeast and East Asians not only manifested genomic mixture with each other, but also preserved their characteristic genetic gradient through evolution [62,63]. Assignment of individuals into distinct ancestries is highly challenging for geneticists, partially due to the lack of clearly defined ancestry variables across studies. Recently, principal component analysis (PCA) methods have been shown to be an accurate and powerful tool to predict the geographical origin of individuals. Spatial Ancestry Analysis (SPA), which explicitly models allele frequencies in European cohorts, has also recently attracted the attention of geneticists [64,65,66]. These approaches have been developed to help researchers accurately stratify study populations and account for corresponding geographical differences in genetic studies.

## 2. GWAS for Fat Distribution in European Populations

### 2.1. Loci Associated with WHR

In 2009, Lindgren et al. performed a meta-analysis including 16 GWAS for WC and WHR [67]. They selected 26 SNPs for follow-up and identified three loci influencing body fat distributions, two loci associated with WC and one WHR locus.

In 2010, Heid et al. performed a large-scale two-stage GWAS for WHR in European populations. They reported 14 loci reaching a genome-wide significance level for association with WHR [19]. As expected, the 14 WHR-associated alleles showed directionally consistent enrichment with increased triglycerides, low-density lipoprotein-cholesterol, fasting insulin and HOMA-derived insulin resistance measures. For example, rs10195252, located in growth factor receptor-bound protein 14 (*GRB14*)-Cordon-bleu WH2 repeat protein-like 1 (*COBLL1*) and associated with higher WHR was also related to increased triglycerides (*p* = 7.4 × 10^−9^), fasting insulin levels (*p* = 5 × 10^−6^) and insulin resistance (*p* = 1.9 × 10^−6^). Eleven of the 14 WHR-related loci showed consistent associations with T2D, of which SNPs in disintegrin-like and metalloprotease with thrombospondin type 1 motif 9 (*ADAMTS9*), nischarin (*NISCH*)-Stabilin-1 (*STAB1*) and inositol 1,4,5-trisphosphate receptor type 2 (*ITPR2*)-Sarcospan (*SSPN*) genes reached nominal significance thresholds of *p* < 0.05. Shungin et al. performed a FD GWAS, which included 210,088 European individuals [20]. The study identified 48 WHR adjusted for BMI loci, of which 33 were novel. To this point, it was the largest study to highlight the role of FD-associated loci in biological processes related to adipogenesis, angiogenesis and insulin resistance, thus strongly supporting the previous findings by Heid et al. [19]. A genome-wide interaction meta-analysis study for adult body size and shape reported 65 SNPs of WHR adjusted for BMI reaching a genome-wide association level [21]. The largest genetic studies on fat distribution so far are based on UK Biobank data, a large-scale biomedical database and research resource, containing in-depth genetic and health information from half a million UK participants [68]. A recent GWAS with this dataset included 450,000 European individuals and identified 202 WHR-related SNPs adjusted for analyzing the association of genetic variants related to different fat depots (gluteofemoral and abdominal). Subsequently, they applied four polygenic scores which were derived using these 202 genetic variants to estimate body fat distribution association with cardiometabolic risk. These four polygenic scores included the overall score of 202 variants, the hip-specific score of 22 variants, the waist-specific score of 36 variants and the overall score of 144 remaining variants. The 202 variant and 144 variant overall polygenic scores were associated with higher abdominal visceral fat mass, lower gluteofemoral fat mass and higher odds of T2D and coronary heart disease. The waist-specific polygenic score was only associated with higher abdominal fat mass, but not with gluteofemoral or leg fat mass. The hip-specific polygenic score was only associated with lower gluteofemoral and leg fat mass, and not with abdominal fat mass. Both waist-specific and hip-specific scores were associated with triglyceride levels, increased risk of T2D and coronary heart disease. In addition, the hip-specific polygenic score was associated with higher fasting insulin and higher low-density lipoprotein levels [23]. This study re-emphasized the role of visceral fat accumulation in cardiovascular and metabolic diseases. It also provided novel evidence for a possible independent role of impaired gluteofemoral fat distribution on cardiometabolic health outcomes. The second large-scale study included nearly 700,000 European subjects and identified 346 BMI-independent WHR loci, of which 300 were novel and contained 463 independent WHR-related signals [24]. Despite the impressive statistical power of these studies, WHR- or WC-associated SNP markers collectively explained only ~4% of the observed variability. In summary, the large proportion of unexplained variability in body fat distribution represents the major limitation in our progress to better understand the genetics of body fat distribution [23,24].

### 2.2. Loci Associated with WC and Hip

It is noteworthy that waist and hip circumference are central measures in the composition of body size and shape and are therefore often considered individually in addition to the WHR, a derivate of these two measures. To date, 73 WC and 63 hip-associated loci independent of BMI but overlap with known WHR loci have been described. These loci seem to be involved in biological processes similar to those observed for WHR loci, thus WC and hip represent further valuable tools for characterizing central obesity [20]. In this context, Tachmazidou et al. assessed the contribution of 15,000,000 sequence variants to 12 anthropometric traits using a hybrid approach of cohort-wide low-depth whole-genome sequencing [69]. They confirmed 106 novel anthropometric variants, of which 12 were new WC variants and seven were hip-related loci.

Main findings from FD-related GWAS conducted in European cohorts are summarized in Table 1.

## 3. GWAS for Fat Distribution in Asian Populations

### 3.1. Loci Associated with WHR

In 2009, the first two-stage GWAS for anthropometric traits in East Asian populations identified WHR related susceptibility locus, which mapped within the HECT domain E3 ubiquitin protein ligase 4 (*HECTD4*) locus [27]. This gene is suggested to play a role in obesity and inflammation [70]. The locus is also associated with thoracic–hip circumference ratio (THR) which was described as a predictor of T2D, but it does not seem to have an impact on WHR in recent Korean GWAS [71]. In 2016, Wen et al. reported another large-scale, two-stage GWAS which revealed two novel loci mapped within Nidogen 2 (*NID2*) and HLA class II histocompatibility antigen DRB5 β chain (*HLA-DRB5*); both associated with WHR adjusted for BMI [26]. The function of *NID2* is unknown, but there is some evidence for its role in adipogenesis [72]. *HLA-DRB5* seems to be associated with islet autoantibodies and risk for childhood type 1 diabetes [73]. Scott et al. conducted a South-Asian-specific GWAS (N = 10,318) and exome-wide association (N = 2637) for WHR, which resulted in replication of some of the previously reported association signals from European studies, but did not reveal any novel loci related to WHR. Only 34 out of 48 WHR previously reported loci showed directionally consistent effects on WHR in South Asians, although most of them provided only poor evidence for a relationship with WHR. Indeed, merely four loci showed a nominally significant effect in South Asians, suggesting that loci identified in European GWAS do not explain the high WHR ratio and the risk of central obesity in the Asian population [28]. The authors proposed (a) European-specific effects of the variants, (b) stronger relationships between tag and causal SNPs in Europeans, (c) winner′s curse in European discovery, and (d) greater phenotypic heterogeneity in South Asians (smaller stature but greater central adiposity than Europeans) as possible explanations for these inconsistencies.

In 2018, Wu et al. conducted a bivariate GWAS including 140 Asian dizygotic twins and analyzed the associations between loci and BMI-WHR in 139 dizygotic twin pairs [74]. They also conducted a heritability analysis in 242 monozygotic and 140 dizygotic twins in Asian populations using the structure equation Cholesky decomposition model. The study showed a modest genetic correlation (r = 0.53) between BMI and WHR. Moreover, it reported 291 nominally associated pleiotropic loci (*p* < 0.05), but none of them reaching significant genome-wide associations. In the bivariate GWAS discovery stage, 26 variants were associated with BMI–WHR jointly variates (*p* < 10^−5^). This study pointed to interconnected pathophysiological networks for a spectrum of BMI and fat distribution. However, one of the limitations of this work is that it failed to replicate the results from better-powered, large-scale studies from Asians and Europeans [20,22,25].

### 3.2. Loci Associated with WC

In 2008, Chambers et al. performed a two-stage GWAS for insulin resistance and related-FD traits in Indian Asians [75]. They confirmed the established association of rs12970134 SNP near the melanocortin 4 receptor (*MC4R*) with WC and proposed the variant might affect mRNA expression of *MC4R*, one of the most prominent genes for monogenic obesity, in the hypothalamus and other parts of the central nervous system. In 2016, Wen et al. reported seven WC (or WC adjusted for BMI)-associated loci in the East Asian population [26]. They found a substantial overlap between loci associated with WC and BMI, which was due to the strong correlation between the two obesity measures. Three of the seven WC-related loci were novel in East Asians and were close to centrosomal protein 120 (*CEP120*), TSC22 domain family member 2 (*TSC22D2*), and solute carrier family 22 member 2 (*SLC22A2*). Four of the seven loci were only associated with WC adjusted for BMI. Two of them (near EGF containing fibulin extracellular matrix protein 1 (*EFEMP1*) and ADAMTS like 3 (*ADAMTSL3*)) have been previously reported in both European and East Asian studies [26,69]. The others were located near the GNAS Complex Locus gene (*GNAS*) and Canopy FGF signaling regulator 2 (*CNPY2*) and independently found in Asian populations. *EFEMP1*, *ADAMTSL3* and *CEP120* were previously reported to affect height as well [76,77], suggesting that these loci may be related to general body size.

## 4. GWAS for Fat Distribution in African Populations

### Loci Associated with WHR and WC

In 2013, Liu et al. performed a collaborative meta-analysis of waist-based traits in African ancestry individuals [78]. The study discovered two novel loci (rs2075064, rs6931262) associated with FD in African people. The WC-related rs2075064 maps to the LIM homeobox 2 (*LHX2*) gene, which was proposed to be related to hair color in people of European ancestry in a recent study. The WHR-associated rs6931262 maps to Ras Responsive Element Binding Protein 1 (*RREB1*) [79], playing a role in the Ras signaling pathway currently known to be related to WHR in European GWAS [36]. Liu et al. also evaluated 14 WHR-related loci previously reported by the GIANT consortium in European populations [19,78]. Six out of the 14 loci showed nominal evidence for association with FD measures such as waist.

In another work by Liu et al., a multi-ethnic fine-mapping of 14 WHR loci was performed using the Bayesian approach that takes advantage of allelic heterogeneity across populations to combine meta-analysis results [29]. Twelve of 14 loci identified in previous European studies remained strongly associated with WHR in combined analyses integrating multiple populations (Europeans and African Americans). Moreover, five loci substantially narrowed the signals to smaller sets of variants compared to previous findings in European studies.

In 2017, in a population of African descent, a fine-mapping study using Metabochip array designed for fine-mapping cardiovascular-associated loci confirmed four out of seventeen previously identified genomic regions associated with WHR in Europeans [30]. It suggests that some of the biological pathways affecting WHR are shared by different ancestries.

Altogether, most FD-associated loci initially discovered in studies including participants of European descent have not been widely replicated in Asian and African individuals so far. Although the majority of previous studies focused on European cohorts [19,20,23,24], recently emerging non-European studies proposed several novel FD susceptibility loci. Some of them showed consistent effects of FD loci across populations, suggesting that FD loci might be partially shared between different ethnicities (Figure 1). In summary, only 22 loci have been found independently in the Asian studies and seven loci in the African studies with a relatively weak threshold *p* < 5 × 10^−6^.

Table 2 highlights 10 FD-related genetic studies conducted in non-European cohorts (Asians and Africans).

## 5. Sexual Dimorphism of Fat Distribution Loci in Different Ethnicities

Adipose tissue is more likely to accumulate in the central depot in men, whereas in women it is preferentially deposited in peripheral areas [80]. This regulatory effect is generally attributed to sex-specific hormonal, environmental and nutritional factors [81]. Notably, the heritability of fat distribution is highly sex-specific as well. For example, the heritability of WHR varied significantly between sexes in the Framingham Heart Study (h^2^_women_ = 0.46, h^2^_men_ = 0.19, P_difference_ = 0.0037) and the TwinGene Study (h^2^_women_ = 0.56, h^2^_men_ = 0.32, P_difference_ = 0.001) [20]. In line with this, numerous genetic studies have found a considerable number of WHR-related loci manifesting marked sexual dimorphism [19,20,21].

### 5.1. Sexual Dimorphism of FD Loci in Europeans

Sex-stratified analyses by Heid et al. showed that 12 of 14 WHR-associated loci reached genome-wide significance level in women, but only three in men [19]. This study with 108,979 women and 82,483 men revealed seven sex-specific genetic loci in R-spondin 3 (*RSPO3*), vascular endothelial growth factor A (*VEGFA*), *GRB14*, lysophospholipase-like 1 (*LYPLAL1*), homeobox C 13 (*HOXC13*), *ITPR2-SSPN* and *ADAMTS9* genes, all with a more substantial effect on WHR in women. Subsequently, Shungin et al. carried out a GWAS including 224,459 individuals and discovered 40 WHR-related loci, 20 of which were sex-specific (19 for women and one for men) [20]. This study supported previous work by Heid et al. by highlighting the strong diversity of the effects of WHR loci on fat distribution according to the respective sex [19]. In a recent meta-analysis, Winkler et al. revealed 44 sex-specific WHR loci adjusted for BMI (27 previously established, 17 novel), of which 28 manifested greater effect in women and five in men [21]. In summary, strong sexual dimorphism is a feature of genetic determinants of body fat distribution, but not of overall obesity as assessed by BMI.

### 5.2. Sexual Dimorphism of FD Loci in Asian

In 2016, a meta-analysis of GWAS for WHR and WC in individuals of East Asian ancestry identified nine novel loci associated with FD in East Asians [26]. Two of them (near *EFEMP1* and *NID2)* showed significant sex differences (P for homogeneity test <  0.05); the SNP rs3791679 (near *EFEMP1*) associated with WC (adjusted for BMI) showed stronger effects in men (effect size: 4.04 vs. 2.43 in women), whereas rs1982963 (near *NID2*) associated with WHR (adjusted for BMI) showed markedly stronger effects in women (effect size: 6.26 vs. 2.88 in men).

### 5.3. Sexual Dimorphism of FD Loci in Africans

To date, only the rs13389219 (near *GRB14*) and rs2059092 (near *ADAMTS9*) loci showed evidence of FD-related sexual dimorphism in Africans by having a stronger effect size in women compared to men [30,78].

In summary, the current knowledge indicates that sexual dimorphism in fat distribution might be at least in part influenced by genetic determinants.

## 6. Potential Regulatory Genes for Ectopic Fat Deposition

Ectopic adipose tissue deposition is strongly associated with the risk for metabolism diseases, including insulin resistance [82], dyslipidemia [83] and coronary heart disease [84,85]. Taking into account the fact that body fat distribution is highly heritable with an estimated range of 36–47% [86], the existence of unique and independent genetic loci for ectopic adipose tissue is not surprising [87,88,89].

In 2011, Foster et al. performed a GWAS of renal sinus fat in the Framingham Heart Study. Although the study was underpowered to detect loci associated with the renal sinus fat at a genome-wide significance level, it discovered 20 independent SNPs related to the renal sinus fat with *p* < 5 × 10^−5^ [88]. In 2012, Fox et al. conducted a pericardial fat GWAS in 5487 subjects of European ancestries and identified a pericardial fat-related locus (independent of overall and visceral obesity) with the variant rs10198628, which maps close to tribbles pseudokinase 2 (*TRIB2*) [89]. This finding supports the concept of a unique genetic basis for ectopic fat distribution. A recent multi-ethnic GWAS meta-analysis established seven loci associated with pericardial adipose tissue area (PAT) near ataxin (*ATXN1*), ubiquitin conjugating enzyme E2E2 (*UBE2E2*), EBF transcription factor 1 (*EBF1), RREB1*, gasdermin B (*GSDMB*), GRAM domain containing 3 (*GRAMD3*) and endosulfine α (*ENSA*) [87]. Among them, *ATXN1* and *UBE2E2* showed a potential functional role in adipocyte differentiation. Knockdown of both genes in subcutaneous adipose tissue (SAT) adipocytes impaired the formation of lipid droplets during adipogenesis, whereas in visceral adipose tissue (VAT), only *Ube2e2* knockdown impaired adipogenesis [87]. *UBE2E2*, an established T2D susceptibility candidate gene, encodes the ubiquitin-conjugating enzyme E2E2 and is expressed in the human pancreas, liver, muscle and adipose tissue [90]. Unlike *ATXN1* and *UBE2E2* which are associated with the PAT area, SAT, VAT, *EBF1* and *ENSA* were only associated with PAT area size, underlining their specificity for this specific fat depot.

Ectopic fat accumulation is the result of complex molecular and cellular mechanisms and each fat depot can be considered an independent endocrine organ. Consequently, their effects on metabolism vary significantly according to the characteristic phenotype of each adipose tissue depot. For instance, SAT adipocytes predominantly produce adiponectin, and VAT adipocytes more actively synthesize leptin [91]. Epicardial adipocytes have high pro-inflammatory activity, whereas most perivascular adipocytes do not synthesize TNF-α [91]. Due to the complex genetic architecture combined with a strong environmental component, ectopic fat accumulation may manifest distinct features in different populations. This heterogeneity in ectopic fat deposition between populations may at least partially explain the observed differences in the risk of obesity comorbidities among populations. Therefore, identifying and developing a better understanding of the unique population-specific fat distribution loci may ultimately help us effectively control the risks of numerous metabolic disorders including T2D, cardiovascular diseases and many others. Extensive research on ectopic adipose tissue could lead to a more accurate stratification of obesity into categories according to adipose tissue distribution, which would significantly improve clinical risk assessments and, subsequently, personalized treatments of patients with obesity.

## 7. Molecular Mechanisms Underlying Trans-Population Differences in Fat Distribution

As mentioned above, several of the FD-related loci initially discovered in European studies have been replicated in Asian and African cohorts (Appendix A). Reviewing all current GWAS related to FD, there were 95 loci that were included in replication efforts in Asians and Africans. Only five loci that reached significant p values in the GWAS and with directionally consistent effects on FD traits were shared among European and Asian populations, whereas no common locus was found between non-African (European, Asian) and African cohorts (Appendix A). According to the defined criteria of *p* < 5 × 10^−8^ for GWAS and *p* < 0.05 for replication, we found 24 out of the 95 FD-related loci being previously reported in European populations and which can be considered as replicated in Asian or African populations. These included 17 loci in Asians and eight loci in Africans, with only one commonly associated locus found in all cohorts (Appendix A). The remaining 71 loci could not be replicated, as demonstrated by the following constellations: (I) Four out of the 71 loci with significant *p*-values based on GWAS (*p* < 5 × 10^−8^) in non-European (three in Asians and one in Africans) were not found in European cohorts, despite adequate statistical power. Several reasons may explain the inconsistencies: (a) different measures of body FD (e.g., WHR adjusted for BMI in European vs. WC in Asian cohorts), (b) different analytical procedures (e.g., no BMI adjustment in Asian population), and (c) genetic heterogeneity. (II) Forty-two of 71 loci reached a genome-wide significant p value (*p* < 5 × 10^−8^) in Europeans, but no significant association (*p* > 0.05) was found in non-Europeans (South Asians and Africans). When analyzed in European cohorts, 29 of the 42 loci were directionally consistent but not significantly associated with FD-related traits in South Asians. Minor allele frequencies are unlikely to be a crucial factor explaining the missing replications because there are no systematic differences in risk allele frequencies compared to European populations [28]. Regarding the comparison between European vs. African cohorts, 11 of 42 variants showed no significant associations in African cohorts. Limited statistical power due to the small sample sizes in African cohorts may have contributed to the lack of statistically significant associations (Appendix A). Another possibility to consider could be the existence of different causal variants between Asian, African, and European individuals, resulting from LD pattern dissimilarities in Asian and African cohorts due to different haplotype structures, leading to population-specific correlations between causal variants and marker SNPs. (III) For the remaining 25 variants, there is no convincing evidence for genome-wide significant associations with FD-related traits in any of the studied populations.

Taken together, only a few FD-related variants have been successfully replicated in non-European cohorts. As mentioned above, the most likely explanation for replication failure is a small sample size, which leads to limited statistical power to detect associations between variants and phenotypes. Moreover, the index variants may differ among populations due to variations in their haplotype structures.

Five FD-related loci shared between East Asian and European ancestry reached statistically significant associations with *p* < 5 × 10^−8^ in both groups (Figure 2). Despite the different allele frequencies, the identified loci have comparable effect sizes on FD in both populations. For example, the MAF of SNP rs3791679, located in *EFEMP1*, is three times higher in East Asians than in Europeans (MAF_Asi_ 0.77 vs. MAF_Eur_ 0.23); however, the effect size is approximately the same (ß_Asi_ 0.031 vs. ß_Eur_ 0.035).

Due to the limited number of large-scale genetic studies with reasonable statistical power in non-European cohort studies, most of the reported loci could not reach the conservative significance threshold of *p* < 5 × 10^−8^ widely used in GWAS. Therefore, for the purpose of this review, we considered all FD-associated loci in Asians and Africans with a less stringent threshold of *p* < 5 × 10^−6^ (Table 3).

We found 22 and 7 FD-related susceptibility loci in Asian and African cohorts, respectively, by screening associations with *p* < 10^−6^ and after pruning by linkage disequilibrium (LD) testing through LD proxy module in the National Institutes of Health online public LD database [94]. These variants associated at the GWAS level in Asian cohorts explained 0.104% of WHR (including WHR adjusted for BMI) and 0.279% of WC (including WC adjusted for BMI) variance. Only one variant with GWAS significance level for WC (adjusted for BMI) explained 0.067% of the phenotypic variance in African studies. These FD-driving loci may harbor promising target genes contributing to a better understanding of fat distribution in multi-populations. The biological functions of these genes may shed light on mechanisms underlying the heterogeneity of fat distribution across populations.

One of the prominent East Asian candidate genes related to the WHR locus (rs1982963) is *NID2*, which encodes a member of the nidogen family of basement membrane proteins as an isoform of nidogen-1 [95]. NID2 protein is a matrix protein that serves as a link between laminin-1 and collagen type IV and thus stabilizes certain basement membranes in vivo and plays an important role in embryogenesis [96]. NID2 and NID1 show a similar distribution in each organ during embryonic development, and NID1 deficiency in the cell is compensated by NID2 [97]. In current research, the adipokine tartrate-resistant acid phosphatase is co-localized with *NID2* in adipose tissue and interacts with the G3 domain of NID2 in pre-adipocytes [72]. It is suggested that NID2 may play an essential role in adipogenesis and consequently in adipose accumulation. A recent GWAS for T2D reported that the highest association hit in the BMI-adjusted model was near *NID2*, which is associated with FD or lipodystrophy traits and was identified only in East Asian individuals [98]. The authors propose that different fat depots play an important role in insulin resistance and T2D among East Asian populations.

The WHR-related locus rs2074356 maps close to *HECTD4*, whose biological function is still largely unknown. It has been recognized that it may function as an E3 ubiquitin-protein ligase, which accepts ubiquitin from an E2 ubiquitin-conjugating enzyme in the thioester form and then directly transfers ubiquitin to target substrates [99]. A Korean study found that homozygosity for the minor alleles of two SNPs in *HECTD4* could reduce the risk of T2D in never-drinkers [100]. One possible mechanism is that *HECTD4*, as E3 ubiquitin-protein ligase, is involved in the ubiquitination of syntaxin 8 [101], and that overexpression in VAT is closely linked to the presence of T2D in patients with obesity [102]. *HECTD4* was also shown to be related to THR in Asian adult men. Notably, THR is one of several anthropometric markers of T2D risk, independent of BMI and WHR in Korean adults [103]. The function of the predicted *HECTD4* transcript has not been well established to date. This gene may be a potential target to explain ethnic differences in fat depot form; however, further functional molecular studies are needed.

Rs2057291 near *GNAS* is associated with WC (adjusted for BMI) in East Asians. *GNAS* encodes the α-subunit of heterotrimeric Gs protein (Gsα) that couples multiple receptors to adenylyl cyclase and is thus required for receptor-mediated stimulation of cellular adenosine 3′, 5′-cyclic monophosphate (cAMP) generation. It is a key component of adenylyl cyclase signal transduction pathways regulated by G-protein-coupled receptors [104]. *Gnas*-deficient mice have a reduced number of β and increased number of α cells in pancreatic islets [105]. A higher level of *GNAS* expression was observed in human islets compared with fat, liver and muscle tissues. A recent study indicated that *GNAS* is a vital gene for the insulin-secreting capacity of β-cells [106]. When *GNAS* is silenced in INS-1 cells, the amount of insulin secretion, insulin content and cAMP production are markedly decreased. Moreover, *Gnas*-knockout mice showed decreased skeletal muscle mass and impaired glucose tolerance [107], and *Gnas*-knockdown in adipocytes in vitro affected the process of adipogenesis [108]. Therefore, *GNAS* does not only affect islet cell function, but also plays a crucial role in adipogenesis.

Interestingly, *SLC22A2*-rs368123, which is related to WC, encodes the organic cation transporter, and there is no evidence for *SLC22A2* being directly related to obesity and diabetes. However, it mediates the active excretion of metformin in the kidney. Available data suggest that variants in the locus affect insulin resistance and β-cell activity in patients with T2D by altering in vivo metformin transport and renal tubular clearance [109,110].

In summary, the biological function of some potential candidate regulatory genes within FD-associated loci (*p* < 5 × 10^−8^) in Asians and Africans remains unknown. For example, the molecular functional mechanisms of *CEP120*, *TSC22D2*, *Al090771.1* and *LHX2* have not been yet linked to obesity, fat distribution and obesity comorbidities. Nevertheless, the underlying biological functions of these genes may provide further insights into genetic differences in FD among non-European and European ancestries.

## 8. Conclusions

The main purpose of this review was to summarize recent advances in the genetics of body fat distribution among populations of diverse ancestries since the distribution patterns of stored fat clearly vary between different ethnicities. These differences may place individuals of diverse ancestries genetically predisposed to store fat preferentially in the visceral depot at an increased risk of developing obesity-related metabolic complications. For example, with gradually increasing BMI, the odds of hypertension and diabetes are significantly greater in Asians compared to non-Hispanic whites or blacks and Hispanics [51].

Genetic overlap between ethnicities suggests a universal and essential physiological pathway and bioprocess in the development of obesity. On the other hand, ancestry-specific genetic associations may illuminate differences in susceptibility to obesity or adverse body FD or concomitant metabolic outcomes observed between populations. In the last decade, GWAS on FD measures proved to be the most efficient tool to identify genetic loci, potentially harboring genes controlling FD. The approach also provided a possibility to explain the diversity of obesity and FD in multiple-ethnic cohorts. In addition, the abundance of novel genes/loci whose biological function is still unclear provides a great challenge for researchers in the near future. Previous GWAS for body fat distribution in Europeans highlighted the role of genes involved in biological processes related to the development and regulation of adipose tissue deposition and angiogenesis [20]. In contrast, the FD-associated candidate genes in Asians are more related to embryonic development or growth development. These FD genes, which enrich the biological process of embryonic and post-embryonic development, may suggest specific developmental characteristics that distinguish patterns of fat deposition in early and later life between different ethnic groups. On the other hand, the biological function of some genes associated with FD in Asian populations remains unclear (Table 3). Further investigations into the functions of these genes may help to uncover the different molecular mechanisms explaining the differences in fat distribution.

Furthermore, it is important to systematically investigate sex differences in the genetic architecture of body fat distribution. Although there is a diversity of FD in ethnicity, it is clear that genetic factors in FD are more likely to be sensitive to sex than to ethnicity. For example, Shungin et al. showed 20 sex-specific WHR associated SNPs, 19 of which showed larger effects in women [20]. This was in line with the higher number of loci identified in women, as well as with previously reported findings [19]. The variance component analyses demonstrated a significantly larger heritability (h^2^) of WHR (adjusted for BMI) in women than in men in the Framingham Heart Study (h^2^_women_ = 0.46, h^2^_men_ = 0.19) [20].

The genetics of ectopic fat depots (e.g., liver, heart, muscle or around large vessels) may include many other gene sets independent of anthropometric indicators such as WHR or BMI [87,88,89]. Considering the fact that dysfunctional adipose tissue that is unable to expand through hyperplasia will lead to visceral accumulation and ectopic fat deposition, it could be hypothesized that some ethnic groups (e.g., Asians) with a higher risk of T2D and lower BMI may be more prone to store considerable amounts of fat in visceral or ectopic depots. To date, studies investigating the genetics of ectopic fat loci in regard to potential ethnic differences are lacking. Therefore, to understand the FD heterogeneity in the context of ethnicity, multi-ethnic ectopic fat GWAS are needed in the near future.

In conclusion, there is an ethnic specificity in fat store patterns, which is partly determined by genetics. Multi-ethnic genetics studies are inevitable to better understand complex traits and to improve the risk assessment of different diseases among different races. In addition, a major goal of future research is to find out the still-unknown biological function of these ‘fat distribution genes’, which may ultimately improve our knowledge on the complex etiology of metabolic diseases and lead to revolutionary advances in their treatment, with an intensified focus on personalized medicine.

## Figures and Tables

**Figure 1 genes-12-00841-f001:**
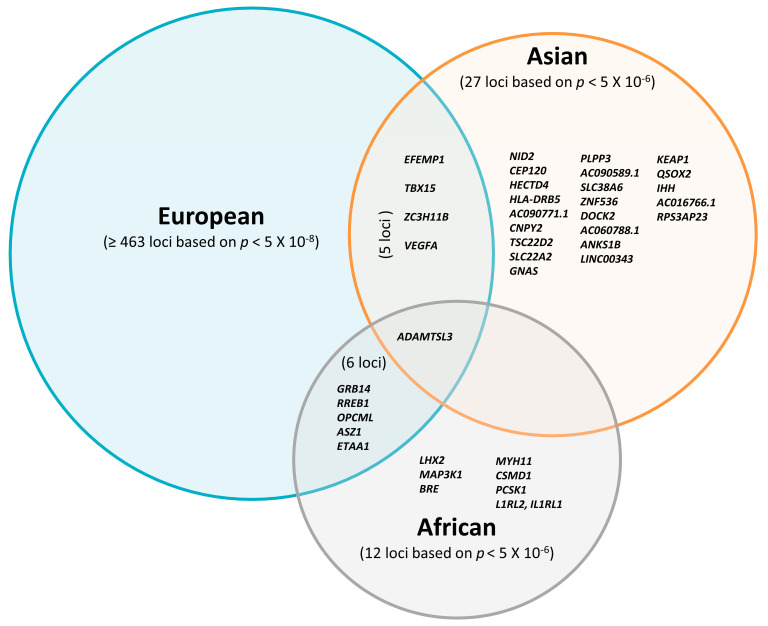
Overlap of reported loci associated with fat distribution traits in Europeans based on *p* < 10^−8^; Asians and Africans are based on *p* < 10^−6^.

**Figure 2 genes-12-00841-f002:**
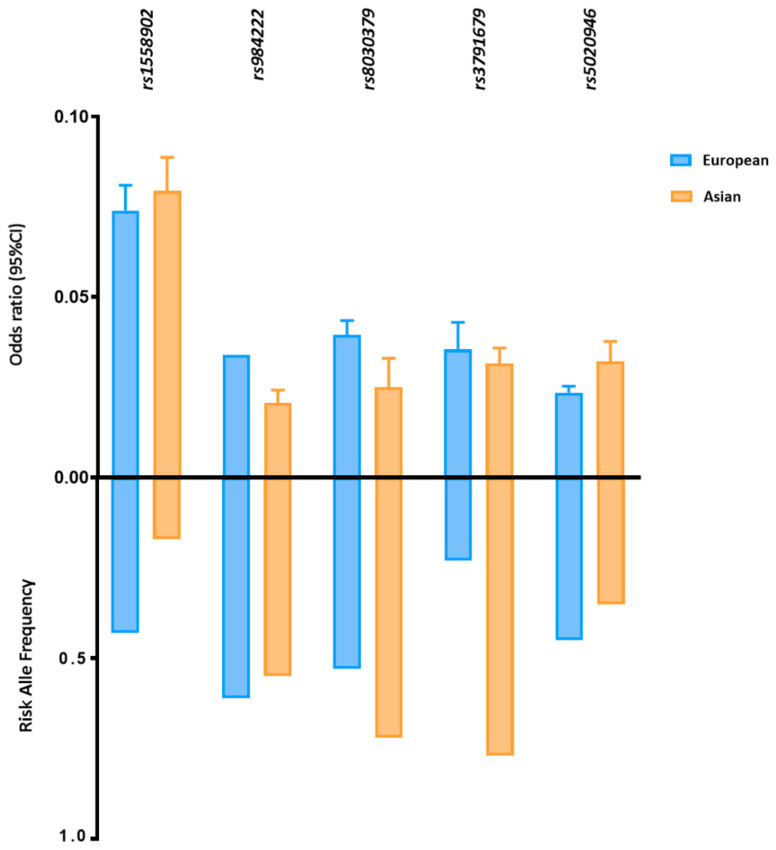
Risk allele frequency and effect size of FD-related loci in European and Asian populations. Risk allele frequency and effect size of fat distribution-related susceptibility loci with genome association significant p value in European and Asian cohorts. Fat distribution-related loci are shown in this figure with a *p* < 5 × 10^−8^ in both cohorts. P values and effect size are according to the reference studies reported in (Appendix A). The allele frequency is based on the 1000 Genomes Project (1KGP).

**Table 1 genes-12-00841-t001:** Studies conducted in a European cohort.

Study Type	Publication Year	Sample Size	Male/Female	Cohort Age Mean (SD)	Traits	Criteria for Discovery ^c^	N° of Variants ^d^ (Discovery Stage)	Criteria for Replication ^e^	N° of Variants ^f^ (Replication Stage)	N° of Sexual Dimorphism Variants	Reference
GWAS	2009	140,644 (Dis ^a^: 38,580 European; Rep ^b^: 102,064 European)	NA	55.73 (9.67)	WC	WC/WHR, *p* < 1 × 10^−5^ BMI *p* > 0.01, Height *p* > 0.005	26	*p* < 5 × 10^−8^	2	NA	[67]
					WHR				1	NA	
Meta-analysis GWAS	2010	190,803 (Dis ^a^: 77,167 European; Rep ^b^: 113,636 European)	82,483/108,979	NA	WHR	*p* < 1.4 × 10^−6^	16	*p* < 5 × 10^−8^	14	7	[19]
Meta-analysis GWAS	2015	210,088 (Dis ^a^: 142,762 European; Rep ^b^: 67,326 European)	95,379/114,709	50 (10.3)	WHRadjBMI	NA	NA	*p* < 5 × 10^−8^	49	20	[20]
Meta-analysis GWAS	2015	320,485 Europeans	51,625/60,654	<=50	WHRadjBMI	*p* < 1 × 10^−5^	NA	*p* < 5 × 10^−8^	44	44	[21]
			90,988/106,622	>50							
Meta-analysis GWAS	2018	663,598 Europeans	206,951/245,351 *	57 (8) *	WHR	*p* < 5 × 10^−8^	202	NA	NA	NA	[23]
		660,648 Europeans			WHRadjBMI	*p* < 5 × 10^−8^					
Meta-analysis GWAS	2019	694,649 Europeans	NA	NA	WHRadjBMI	*p* < 5 × 10^−9^	346	NA	NA	105	[24]
GWAS	2017	239,856 Europeans (Dia ^a^: 33,811, Rep ^b^: 206,045)	123,766/116,090	51.4	WHRadjBMI	*p* < 1 × 10^−5^	85	*p* < 5 × 10^−8^	1	3	[69]
		54,693 Europeans (Dia ^a^: 34,088, Rep ^b^: 206,053)	28,276/26,417	51.4	WHR		80		1	0	
		67,767 Europeans (Dia ^a^: 47,095, Rep ^b^: 206,723)	34,494/33,273	52	WCadjBMI		139		7	3	
		68,267 Europeans (Dia ^a^: 47,593, Rep ^b^: 20,6737)	33,792/34,475	50.1	WC		118		12	5	
		54,595 Europeans (Dia ^a^: 34,004, Rep ^b^: 205,909)	28,008/26,587	51.4	HIP		114		7	2	

NA: Not applicable. * This study participants included two parts: Ukbiobank and GIANT—the radio of sex and age only showed in Ukbiobank participants; ^a^ Discovery stage sample; ^b^ Replication stage sample; ^c^ Criteria for discovery stage; ^d^ Number of variants found in the discovery stage by c criteria; ^e^ Criteria for replication in Asian study (also as a criterion for other types of studies); ^f^ Number of variants found in the replication stage by ^e^ criteria.

**Table 2 genes-12-00841-t002:** Studies conducted in cohorts of Asians and Africans.

Study Type	Publication Year	Sample Size	Male/Female	Cohort Age Mean (SD)	Traits	Criteria for Discovery	N° of Variants ^d^ (Discovery Stage)	Criteria for Replication ^e^	N° of Variants ^f^ (Replication Stage)	N° of Variants Replicated from ^g^	N° of Variants Successfully Replicated from ^g^	N° of Sexual Dimorphism Variants	Reference
**(1) Studies conducted in Asian populations**													
GWAS	2009	16,703 (Dis ^a^: 8842 East Asian; Rep ^b^: 7861 East Asian)	7397/9306	54.4 (8.4)	WHR	*p* < 1.0 × 10^−5^	2	*p* < 5 × 10^−2^	1	NA	NA	NA	[27]
GWAS	2016	73,596 East Asian (Dis: 48,312, Rep: 25,284)	31,570/42,026	54.4 (9)	WHRadjBMI	*p* < 1.0 × 10^−6^	33	*p* < 5 × 10^−8^	2	60	11 (1 × 10^−3^)	3 (10 from h)	[26]
					WHRnoBMI		33	*p* < 5 × 10^−8^	0		13 (1 × 10^−3^)		
	2016	78,336 East Asian (Dis: 53,052, Rep: 25,284)	36,310/42,026	54.4 (9)	WCadjBMI	*p* < 1.0 × 10^−6^	33	*p* < 5 × 10^−8^	4	60	7 (1 × 10^−3^)		
					WCnoBMI		33	*p* < 5 × 10^−8^	3	60	15 (1 × 10^−3^)		
GWAS	2016	12,240 (Dis ^a^: 10,318 South Asian; Rep ^b^: 1922 South Asian)	9825/2415	50.5 (11.2)	WHR	*p* < 5 × 10^−8^ *p* < 5 × 10^−2^	0	*p* < 5 × 10^−2^	0	48	4 (*p* < 0.05)	2 (11from known WHR)	[28]
Exome-Wide Association Study	2016	2637 South Asian	1798/839	51.6 (10.1)	WHR	*p* < 1.5 × 10^−6^ (single variant) *p* < 2.5 × 10^−6^ (gene-based analyses)	0	*p* < 5 × 10^−2^	0	NA	NA	NA	[28]
GWAS	2018	274 East Asian (139 dizygotic twin pairs)	NA	over 30	BMI-WHR	*p* < 1.0 × 10^−5^	26	NA	NA	NA	NA	NA	[74]
GWAS	2008	14,639 (Dis ^a^: 2684 South Asian, Rep ^b^: 7394 South Asian and 4561 European)	9954/4685	51.1 (11)	WC	*p* < 1.0 × 10^−5^	31	*p* < 5 × 10^−7^	4	NA	NA	NA	[75]
**(2) Studies conducted in African populations**													
GWAS	2013	33,738 (Dis ^a^: 23,564, Rep ^b^: 19,744 African American or Afro-Caribbean)	9224/24,514	56.1 (9.5)	WC	*p* < 5 × 10^−6^	25	*p* < 5 × 10^−8^	1	NA	NA	NA	[78]
		27,489 (Dis ^a^: 19,744, Rep ^b^: 7745 African American or Afro-Caribbean)	6446/21,043	55.1 (9.7)	WHR	*p* < 5 × 10^−6^		*p* < 5 × 10^−8^	1	14	6	1	
Fine mapping	2014	19,744 African American	10,318/9426	51.4 (9.4)	WHR	NA	NA	*p* < 5 × 10^−8^	NA	14	12	NA	[29]
Fine mapping	2016	15,981 African American	3884/12,097	54.1 (7.8)	WC	NA	NA	*p* < 9.97 × 10^−5^	NA	17	0	0	[30]
					WHR	NA	NA	*p* < 9.97 × 10^−5^	NA	17	8	5	

NA: Not applicable; ^a^ Discovery stage sample; ^b^ Replication stage sample; ^c^ Criteria for discovery stage; ^d^ Number of variants found in the discovery stage by c criteria; ^e^ Criteria for replication in Asian study (also as a criterion for other types of studies); ^f^ Number of variants found in the replication stage by ^e^ criteria; ^g^ European GWAS including Heid et al. 2010 [19], Shungin et al. 2015 [20].

**Table 3 genes-12-00841-t003:** Fat distribution traits susceptibility loci identified (*p* < 5.0 × 10^−6^) in cohorts of Asian and African ancestries.

SNP	Candidate Gene(s) ^a^	Chr ^b^	Allele ^c^ ALT/REF ^d^	RAF ^e, f^	ß-Estimates ^c^ (SE) ^e^	*p*-Value	Reported Traits ^g^	Reference	Explained Variance (%) ^f^
rs1982963	*NID2*	14	A/G	0.85	0.048 (0.012)	**1.0 × 10^−14^**	WHRadjBMI	[26]	0.059
rs10051787	*CEP120*	5	C/T	0.59	−0.04 (0.012)	**7.0 × 10^−12^**	WC	[26]	0.078
rs2074356	*HECTD4*	12	A/G	0.13	0.006 (0.002)	**8.0 × 10^−12^**	WHR	[27]	0.001
rs5020946	*HLA-DRB5*	6	T/G	0.35	0.031 (0.01)	**1.0 × 10^−9^**	WHRadjBMI	[26]	0.044
rs12970134	*AC090771.1*	18	A/G	0.32	NA	**2.0 × 10^−9^**	WC	[75]	NA
rs3809128	*CNPY2 (AC073896.2)*	12	T/C	0.16	−0.037 (0.012)	**4.0 × 10^−9^**	WCadjBMI	[26]	0.037
rs1868673	*TSC22D2*	3	C/A	0.40	−0.044 (0.016)	**1.0 × 10^−8^**	WC	[26]	0.093
rs368123	*SLC22A2*	6	G/A	0.40	0.032 (0.0129)	**3.0 × 10^−8^**	WC	[26]	0.049
rs2057291	*GNAS*	20	G/A	0.77	0.025 (0.01)	**4.0 × 10^−8^**	WCadjBMI	[26]	0.022
rs4912314	*PLPP3*	1	T/C	0.24	0.029 (0.012)	3.0 × 10^−7^	WHRadjBMI	[26]	NA
rs2025924	*LINC00343*	13	C/T	NA	NA	4.0 × 10^−7^	WHR	[74]	NA
rs3100776	*IHH*	2	C/T	0.47	0.017 (0.011)	4.0 × 10^−7^	WCadjBMI	[26]	NA
rs11103390	*QSOX2*	9	C/T	0.25	0.017 (0.006)	5.0 × 10^−7^	WCadjBMI	[26]	NA
rs1507456	*AC060788.1*	8	C/T	NA	NA	7.0 × 10^−7^	WHR	[74]	NA
rs17197710	*AC090589.1*	11	C/T	0.05	−0.06 (0.024)	8.0 × 10^−7^	WHRadjBMI	[26]	NA
rs12227147	*ANKS1B*	12	A/T	NA	NA	1.0 × 10^−6^	WHR	[74]	NA
rs139256956	*ZNF536*	19	C/A	0.03	−0.25 (0.1)	1.0 × 10^−6^	WHRadjBMI	[28]	NA
rs57561811	*SLC38A6*	14	C/T	0.25	−0.07 (0.02)	2.0 × 10^−6^	WHRadjBMI	[28]	NA
rs35316183	*DOCK2*	5	A/G	NA	NA	3.0 × 10^−6^	WHR	[74]	NA
rs17178527	*RPS3AP23*	6	A/G	NA	NA	3.0 × 10^−6^	WC	[92]	NA
rs79817709	*KEAP1*	19	T/G	NA	NA	5.0 × 10^−6^	WHR	[74]	NA
rs4667458	*AC016766.1*	2	G/A	NA	NA	5.0 × 10^−6^	WC	[92]	NA
**Identified Fat Distribution Susceptibility Loci from African Population Based on *p* < 5 × 10^−6^**									
rs2075064	*LHX2*	9	T/C	0.07	−0.07 (0.01)	**2.2 × 10^−8^**	WCadjBMI	[78]	0.067
rs6867983	*MAP3K1*	5	T/C	0.24	−0.09 (0.02)	1.4 × 10^−7^	WC_men	[78]	NA
rs7601155	*BRE*	2	T/C	0.16	0.06 (0.05)	1.7 × 10^−7^	WCadjBMI	[78]	NA
rs17213965	*MYH11*	16	T/C	0.16	0.12 (0.02)	8.8 × 10^−7^	WHRadjBMI_men	[78]	NA
rs11777345	*CSMD1*	8	G/C	0.08	−0.19 (0.04)	3.2 × 10^−6^	WHRadjBMI_men	[78]	NA
rs1345301	*IL1RL2; IL1RL1*	2	G/A	0.19	−0.08 (0.02)	4.6 × 10^−6^	WC_men	[78]	NA
rs2570467	*PCSK1*	5	G/A	0.11	0.1 (0.02)	1.2 × 10^−6^	WC_men	[78]	NA

NA: Not applicable. ^a^ Predicted gene is reported in reference studies. ^b^ Chromosomes are based on NCBI Build154 (GRCh38). ^c^ Alternative alleles were treated as effective alleles. ^d^ The allele frequency is based on the 1000 Genomes Project (1KGP). ^e^ ‘Standard Error’ according to reference studies reported. ^f^ ‘Explained variance’ is the variance explained by each reported variant using the formula which uses the allele frequency (f) estimated in GWAS and estimates of the additive effect (ß) in meta-analysis: Explained variance = ß^2^ (1-f)2f [93]. To estimate the additive explained variance of 22 newly identified FD-related loci in the Asian population, the explained variance of each individual was combined and was 0.104% of WHR (including WHR adjusted for BMI) and 0.279% of WC (including WC adjusted for BMI). ^g^ The reported traits are according to reference. Bold *p-*value indicates statistical significance at GWAS level.

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
