# Peer review of "Genetics of Body Fat Distribution: Comparative Analyses in Populations with European, Asian and African Ancestries"

_genes, 2021, doi:10.3390/genes12060841_

Round 1
Reviewer 1 Report
Sun et al. in this review summarised the genetics of fat distribution in different populations; specifically, they compare the fat distribution in European, Asian and African populations discussing the role of genetic variants in ancestry. They also discuss the sexual dimorphism and the molecular mechanism underlying these differences in fat distribution. The manuscript is well presented, with a good structure. The authors not just mention a high number of studies but also discuss and conclude really well at the end of each section. But first the authors should address some minor comments:
- In the introduction, there is a repetitive sentence: lines 30-32 and lines 41-42, say the same. I think they should consider re-write to avoid that it sounds repetitive.
- In the same section, lines 47-49 and 51-52, explain that visceral fat is the fat depot that present more metabolic risk in both sentences, sounding again redundant. Combine both sentences.
- Section 1.3, between lines 73-79 repeat the word “heritability” six times; re-write this and use other synonyms would sound better.
- Section 4 has exactly the same title that section 3. I assume, following the content, that the authors wanted to say in section 4 “African populations”. Please clarify this.
- Figure 2: the third column does not have error bar, it this a mistake or it is not shown?
Author Response
"Please see the attachment"

Reviewer 2 Report
This review summarises differences in the genetics of obesity and body fat distribution according to race. Obesity increases the risk of type 2 diabetes, hypertension, and cardiovascular disease. The causes of the racial/ethnic differences in incidence of obesity are not well understood. The review contains interesting data on the problem, and I have no significant comments. There are some mistakes that need to be corrected:
Line 112 - ancestry samples such as Genome-wide Polygenic Score (GPS) (The term Genome-wide Polygenic Score (GPS) is not used, common terms are GRS -Genetic Risk Score or PRS - Polygenic Risk Score)
Line 286 - GWAS for Fat Distribution in Asian Populations (the text describes the African population)
Author Response
"Please see the attachment"
